# Groundwater Level Change Management on Control of Land Subsidence Supported by Borehole Extensometer Compaction Measurements in the Houston-Galveston Region, Texas

**Yi Liu [1,\*], Jiang Li [1] and Zheng N. Fang [2]** 

[1]   Civil Engineering Department, Morgan State University 1700 E Cold Spring Ln, Baltimore, MD 21251, USA; jiang.li@morgan.edu

[2]   Civil Engineering Department, University of Texas at Arlington, 416 Yates St., Arlington, TX 76019, USA; nickfang@uta.edu

\*   Correspondence: yi.liu@morgan.edu; Tel.: +1-443-885-3067

**Abstract:** As much as 3.05 m of land subsidence was observed in 1979 in the Houston-Galveston region as a result primarily of inelastic compaction of aquitards in the Chicot and Evangeline aquifers between 1937 and 1979. The preconsolidation pressure heads for aquitards within these two aquifers were continuously updated in response to lowering groundwater levels, which in turn was caused by continuously increasing groundwater withdrawal rates from 0.57 to 4.28 million $m^3$/day. This land subsidence occurred without any management of changes in groundwater levels. However, the management of recovering groundwater levels from 1979 to 2000 successfully decreased inelastic compaction from about 40 mm/yr in the early 1980s to zero around 2000 through decreasing groundwater withdrawal rates from 4.3 to 3.0 million $m^3$/day. The inelastic consolidation that had existed for about 63 years roughly from 1937 to 2000 caused a land subsidence hazard in this region. Some rebounding of the land surface was achieved from groundwater level recovering management. It is found in this paper that subsidence of 0.08 to 8.49 mm/yr owing to a pseudo-constant secondary consolidation rate emerged or tended to emerge at 13 borehole extensometer station locations while the groundwater levels in the two aquifers were being managed. It is considered to remain stable in trend since 2000. The subsidence due to the secondary consolidation is beyond the control of any groundwater level change management schemes because it is caused by geo-historical overburden pressure on the two aquifers. The compaction measurements collected from the 13 extensometers since 1971 not only successfully corroborate the need for groundwater level change management in controlling land subsidence but also yield the first empirical findings of the occurrence of secondary consolidation subsidence in the Quaternary and Tertiary aquifer systems in the Houston-Galveston region.

**Keywords:** borehole extensometer; groundwater withdrawal; groundwater level change management; land subsidence; consolidation; compaction

---

## 1. Introduction

Land subsidence (LS) can be a gradual settling or sudden sinking of the Earth's surface owing to subsurface movement of earth materials [1]. LS is a global problem that has geohazardous impacts on infrastructure and the environment. In the United States, 45 states with more than 44,030 $km^2$ have been directly affected by LS [1]. More than 80 percent of the subsidence in the nation is identified as a consequence of human impact on subsurface water [1,2]. In the early 1900s, the Houston area began to

show the first signs of human-induced LS—initially attributed to extraction of oil and gas from the subsurface alone [3], and has been subsiding due to the combined effects of groundwater withdrawal, hydrocarbon extraction, salt dome movement, and faulting [4]. By 1977, the withdrawals from the Chicot and Evangeline aquifers had resulted in groundwater level declines of 91.5 and 106.75 m below datum in the two aquifers, respectively, in southern and eastern Harris County. Correspondingly, by 1979, as much as 3.05 m of LS had occurred in the Houston-Galveston area, Texas [1]. Approximately, 8,288 $km^2$ had subsided more than 30.5 cm in this region (see Figure 1), which has shifted the position of the coastline and damaged the distribution of wetlands and aquatic vegetation. LS is of particular concern in low-lying coastal areas such as the Houston-Galveston region. Subsidence in the region has increased the frequency and severity of flooding [5]. Low-pressure weather systems, such as tropical storms and hurricanes, result in high rates of precipitation and cause high tides to reach farther inland. Subsidence exacerbates the effects of storm surges and impedes storm water runoff by creating areas of decreased land-surface elevations where water accumulates. The latest flood caused by Hurricane Harvey (2017) in the Houston-Galveston region (HGR) was regarded as one of the costliest disasters in the U.S. history, with damage exceeding $100 billion. Subsidence has shifted the shoreline along Galveston Bay, as evidenced by the inundation of the Brownwood Subdivision associated with Hurricane Alicia in August 1983 near Baytown, Texas, and adjacent areas in the Houston-Galveston region, thereby changing the distribution of wetlands and aquatic vegetation [5]. In the 1950s and 1960s, local area governments began to analyze the serious and very real impact subsidence could have on the area's potential growth and quality of life, and, just as important, began to determine what exactly could be done about it. With a number of studies linking groundwater withdrawal to subsidence—and ongoing measurements confirming those findings—groups of citizens began to work for a reduction in groundwater use in the late 1960s. By 1973, the City of Galveston had begun converting to surface water supplied from Lake Houston, and from 1976 more sources from Trinity River and Brazos River started. The total source of the replenishment was increased from a little bit less than 300 MGD in 1976 to 750 MGD in 2017. In May of 1975, the Texas Legislator created the Harris-Galveston Subsidence District (HGSD), the first of its kind in the United States, to "end subsidence" and to restrict groundwater withdrawal. By 1976, HGSD had begun the process of compiling hydrologic information on the characteristics of the Chicot and Evangeline aquifers, engineering planning information on water usage and water supply in Harris and Galveston counties, and implementing regulatory procedures associated with their first groundwater regulatory plan. USGS measures groundwater levels in over 700 wells in an 11-county area annually in the Houston-Galveston region in order to develop a regional depiction of groundwater levels. The cumulative compaction in the Chicot and Evangeline aquifers are measured at 13 borehole extensometer stations in this region since 1973. The up to 44 years of extensometer compaction measurements, which was established based on the theory of aquifer system compaction due to ground fluid withdrawal with the above practice, can be employed to evaluate the long-term groundwater level change management on control of LS in the Houston-Galveston region, which is the motivation of this paper. It is assumed in this paper that a bulk land subsidence rate is the sum of inelastic and elastic compaction rates due to groundwater withdrawal and secondary consolidation rate due to geo-historical overburden pressure. The 'primary' compaction rate can be zero but the secondary consolidation rate might remain a pseudo-constant during a multi-year period in response to long-term consolidation lasting over 1000 years. The characteristics of these three distinct compaction or consolidation rates that combine to yield a bulk subsidence rate can be distinguished within the bulk borehole extensometer compaction data and groundwater level data when analyzed correctly. The secondary consolidation due to geo-historical overburden pressure and its pseudo-constant rate characteristic found with extensometer compaction data in this paper have not been recognized well in literature other than the primary components in unconsolidated or semi-consolidated aquifer systems in the world. It is showed in this paper that this secondary compaction is uncontrollable rather than primary compaction through long-term groundwater level management.

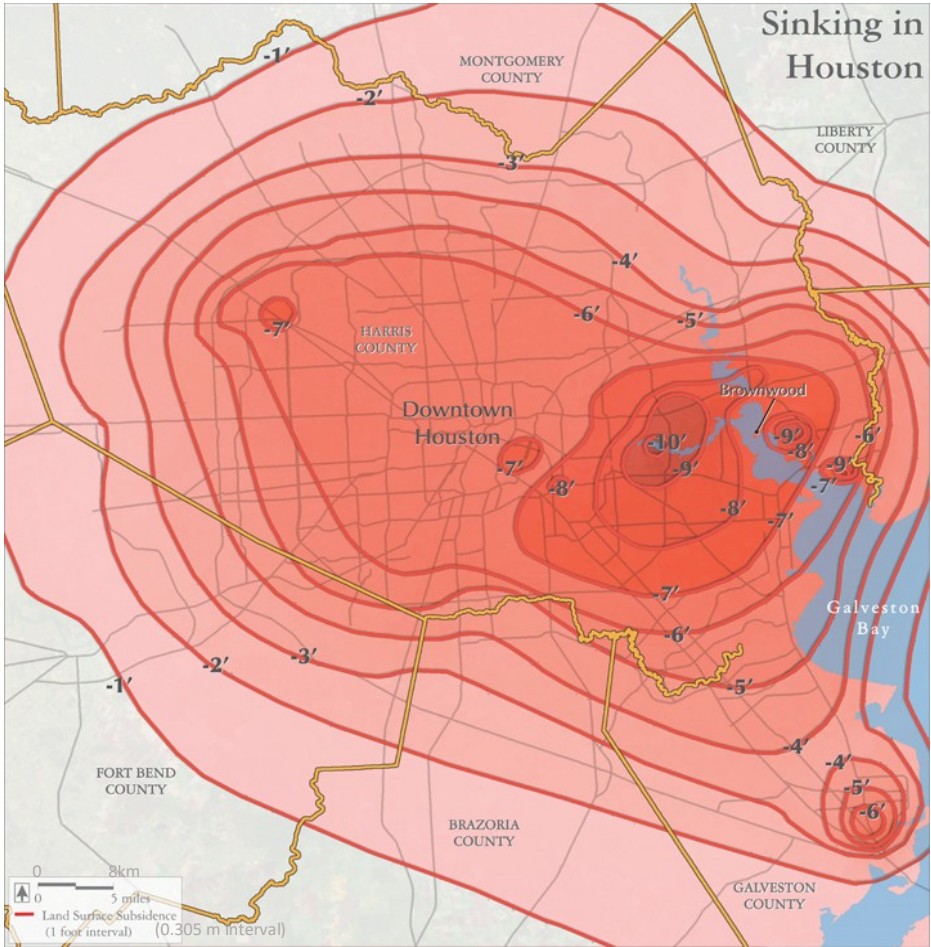

**Figure 1.** The observed land subsidence in the Houston-Galveston region [6], a historical flooding zone frequently hit by hurricanes such as Harvey, Ike, Allison, etc. (−1′ = −.305 m; −2′ = −0.610 m; −3′ = −0.915 m; −4′ = −1.22 m; −5′ = −1.525 m; −6′ = −1.83 m; −7′ = −2.135 m; −8′ = −2.440 m; −9′ = −2.745 m; −10′ = −3.05 m). (Image from TexasLandscapeProject.org; Map sources from U.S. Geological Survey (USGS) [6]).

## 2. Compressible Aquifer System

This section introduces three compressible aquifers and one compressible confining unit that exist in the HGR. Compaction of the aquifer systems was observed with borehole extensometers.

From northwest to southeast, the HGR includes Grimes County with a high elevation of close to 122 m, Montgomery County, Waller County, Harris County, and Galveston County with a low elevation of about 0 to 15 m along the coast of Gulf of Mexico (Figure 2) [7]. The three primary Quaternary and Tertiary aquifers in the Gulf Coast aquifer system in the Houston-Galveston region study area are the Chicot, Evangeline, and Jasper (Figures 2 and 3) [7–10], which are composed of laterally discontinuous deposits of gravel, sand, silt, and clay. The youngest and uppermost Quaternary aquifer, the Chicot aquifer, consists of Holocene- and Pleistocene-age sediments; the underlying Tertiary Evangeline aquifer consists of Pliocene- and Miocene-age sediments; and the oldest and most deeply buried Tertiary aquifer, the Jasper aquifer, consists of Miocene-age sediments (Figures 2 and 3) [5,7]. The lowermost unit of the Gulf Coast Tertiary aquifer system is the Miocene-age Catahoula confining system, which includes Catahoula Sandstone. The Catahoula confining system consists of sands in the upper section and clay and tuff interbedded with sand in the lower section.

Since about 1932, numerous authors have contributed to the body of knowledge and understanding of the complex stratigraphic and hydrogeologic relations of the Gulf Coast aquifer system in the Houston-Galveston study area (Figure 3) [7]. Using this information, a series of groundwater flow models were created, the most recent being by Kasmarek (2013) [6]; these models provide an evaluative tool that can be used by water-resource managers to help regulate and conserve the important natural water resource of the aquifer system and manage groundwater level changes in order to 'end subsidence'.

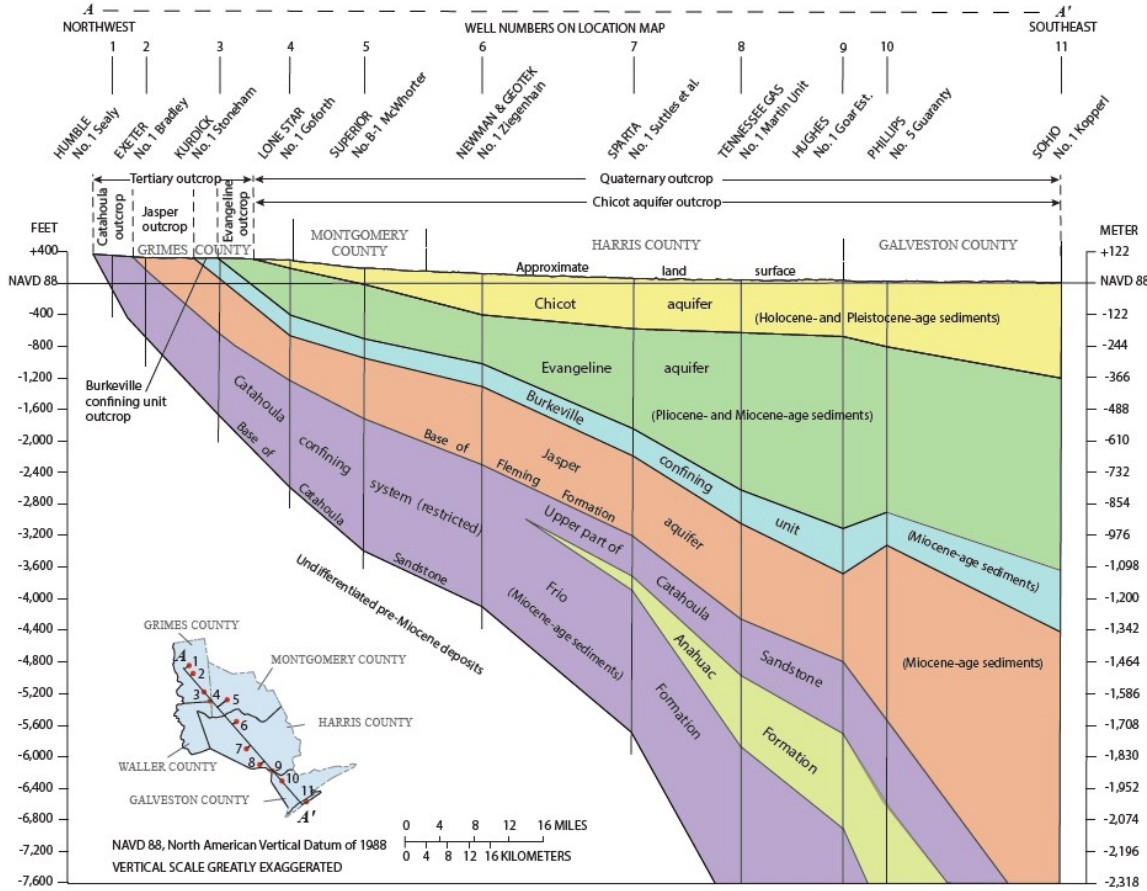

**Figure 2.** Hydrogeologic section A–A′ of the Gulf Coast aquifer system in Grimes, Montgomery, Harris, and Galveston Counties, Texas (modified from Kasmarek et al. (2016) [7] and Baker (1979, 1986) [8,9]). (Map source: USGS [7]).

The percentage of clay and other fine-grained clastic material generally increases with depth downdip [9]. Through time, geologic and hydrologic processes created accretionary sediment wedges (stacked sequences of sediments) more than 2318 m thick at the coast (Figure 2) [11,12]. The sediments composing the Gulf Coast aquifer system were deposited by fluvial–deltaic processes and subsequently were eroded and redeposited (reworked) by worldwide episodic changes in sea level (eustacy) that occurred as a result of oscillations between glacial and interglacial climate conditions [13]. The Gulf Coast aquifer system consists of hydrogeologic units that dip and thicken from northwest to southeast (Figure 2); the aquifers thus crop out in bands inland from and approximately parallel to the coast and become progressively more deeply buried and confined toward the coast [6]. The Burkeville confining unit is stratigraphically positioned between the Evangeline and Jasper aquifers (Figure 2), thereby restricting groundwater flow between the Evangeline and Jasper aquifers. There is no confining unit between the Chicot and Evangeline aquifers; therefore, the aquifers are hydraulically connected, which allows groundwater flow between the aquifers (Figure 2). Because of this hydraulic connection, water-level changes that occur in one aquifer can affect water levels in the adjoining aquifer [14].

Supporting evidence of the interaction of groundwater flow between the Chicot and Evangeline aquifers is demonstrated by comparing the two long-term (1977–2015) water-level-change maps [15], which indicate that the areas where water levels have risen or declined are approximately spatially coincident. Hydraulic properties of the Chicot aquifer do not differ appreciably from the hydrogeologically similar Evangeline aquifer, but can be differentiated on the basis of hydraulic conductivity [16]. From aquifer test data, Meyer and Carr (1979) [16] estimated that the transmissivity of the Chicot aquifer ranges from 915 to 7625 m$^2$/d and that the transmissivity of the Evangeline aquifer ranges from 915 to 4575 m$^2$/d. The Chicot aquifer outcrops and extends inland from the Gulf of Mexico coast and terminates at the most northern updip limit of the aquifer. The recharge rate across the outcrop area ranged from 6.35 mm/yr to 177.8 mm/yr [6]. Proceeding updip and inland of the Chicot aquifer, the older hydrogeologic units of the Evangeline aquifer, the Burkeville confining unit, and the Jasper aquifer sequentially outcrop (Figure 2). In the outcrop and updip areas of the Jasper aquifer, the aquifer can be differentiated from the Evangeline aquifer on the basis of the depths to water below land-surface datum, which are shallower (closer to land surface) in the Jasper aquifer compared to those in the Evangeline aquifer. Additionally, in the downdip parts of the aquifer system, the Jasper aquifer can be differentiated from the Evangeline aquifer on the basis of stratigraphic position relative to the elevation of the Burkeville confining unit (Figure 2).

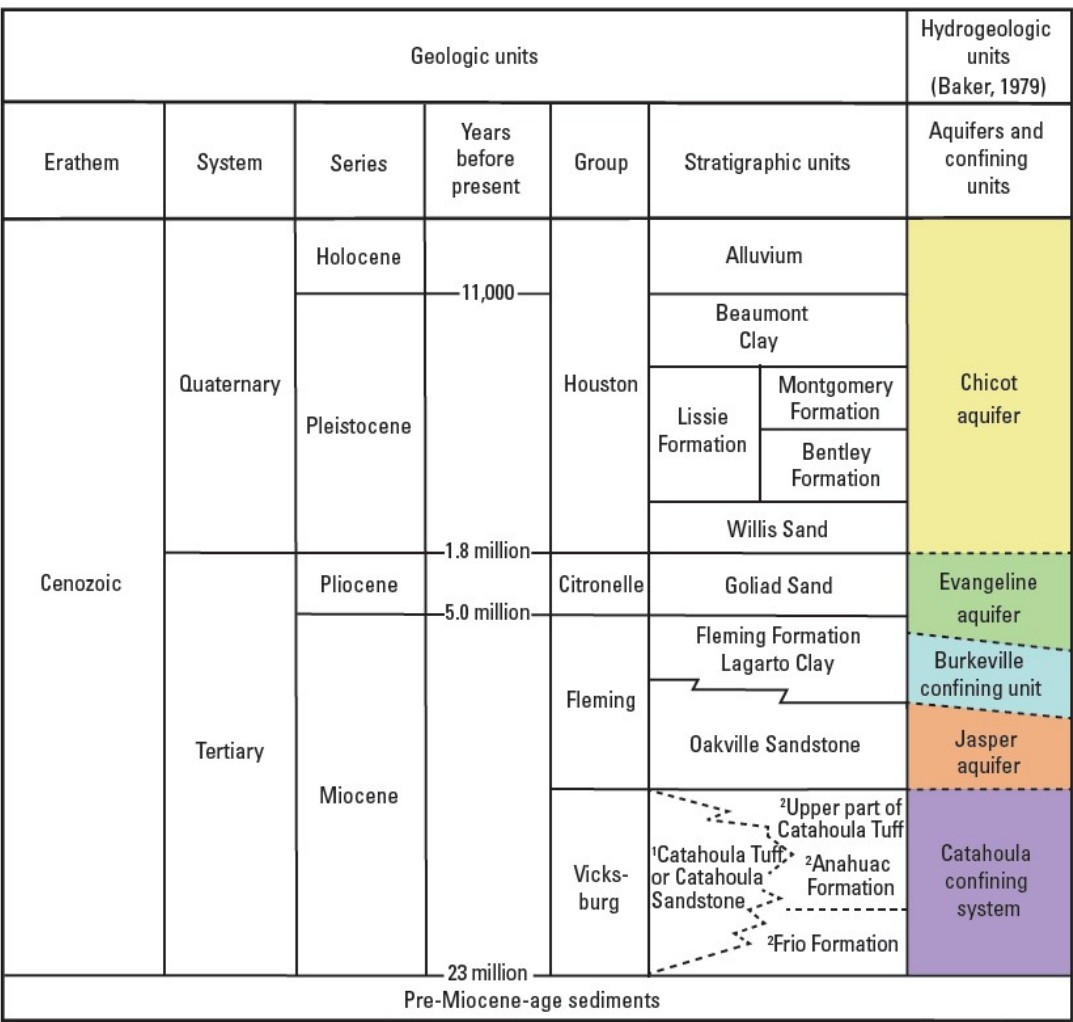

**Figure 3.** Geologic and hydrogeologic units of the Gulf Coast aquifer system in the Houston-Galveston region study area, Texas (modified by Kasmarek et al. (2016) [7] from [8–10]). (Map source: USGS [7]).

## 3. Borehole Extensometer

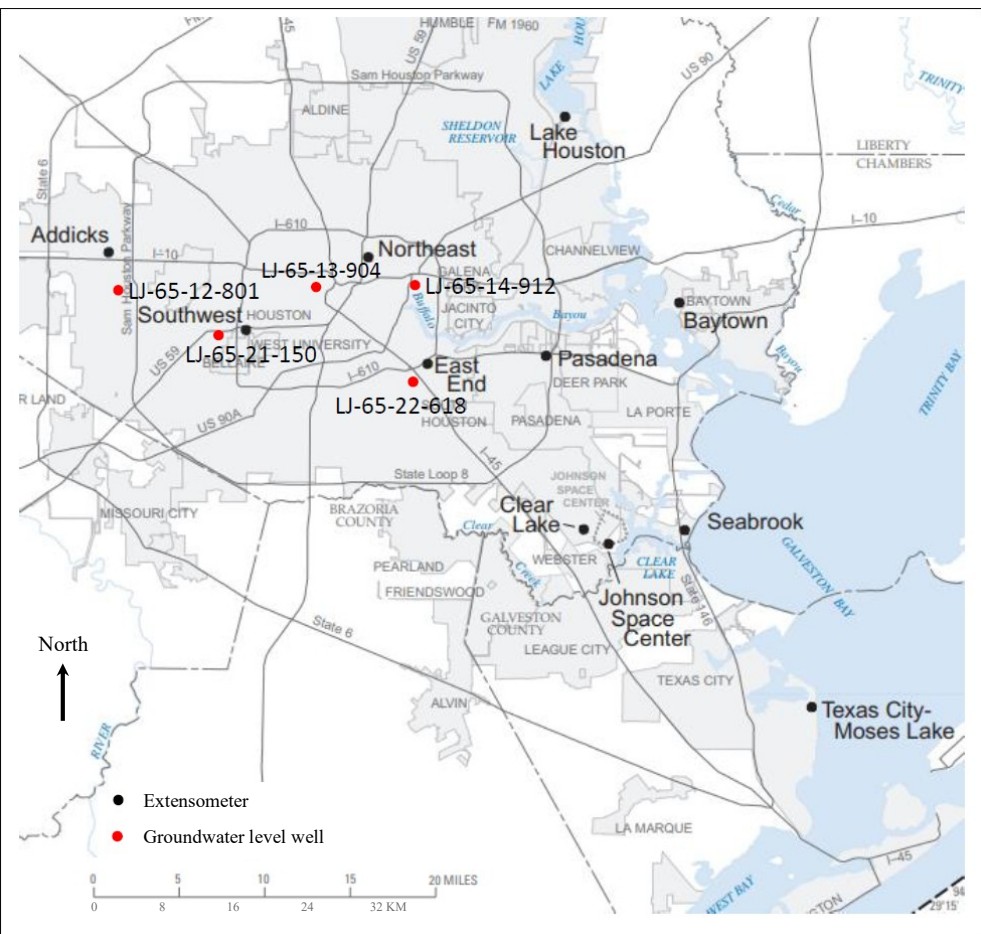

**Figure 4.** Location of borehole extensometer sites and selected groundwater level monitoring well sites, Houston-Galveston region, Texas [17]. (Note: The location of wells LJ-65-21-229 and LJ-65-21-227 is at the same location of extensometer Southwest). (Map source: USGS [17]).

This section introduces the borehole extensometer technology which has successfully measured the compaction of the compressible aquifer systems in the HGR.

Figure 4 shows the 11 borehole extensometer station locations in the Houston-Galveston region [17]. There are two extensometers (shallow and deep) at Baytown and Clear Lake stations, respectively. Each of the other nine stations has only one extensometer. In total, there are 13 extensometers. Figure 5 shows one example of a borehole extensometer at Pasadena in this region. Based on Gabrysch (1984) [18] and Kasmarek et al. (2015) [15], a borehole is first drilled to a predetermined depth, generally below the depth of expected water-level decline in order to construct an extensometer (example shown in Figure 5). A steel outer casing with a slip joint and screened interval is installed in the previously drilled borehole. The slip joint helps to prevent crumpling and collapse of the well casing as compaction of subsurface sediments occurs, while the screened interval allows groundwater to enter the outer casing and inner casing (piezometer) so that the depth to water below land surface can be determined for the aquifer at the depth of the screened interval. A substantial concrete plug is installed and set at the base of the extensometer, and after the concrete plug hardens, the smaller diameter inner pipe (often referred to as the 'extensometer pipe') is inserted down hole inside the outer casing and positioned to rest on the upper surface of the concrete plug at depth. Therefore, this rigid inner pipe extends vertically from the top of the concrete plug to slightly above land surface, thus providing a fixed reference elevation above land surface for measuring changes in land surface elevation. At land surface, a concrete slab is poured and connected to an array of vertical concrete piers extending down

into the water table. The concrete piers connect the slab to the underlying unconsolidated sediments penetrated by the borehole; this construction design helps to eliminate the continuous shrink and swell of the surficial clayey sediments associated with soil-moisture changes. A metal gage house (not depicted in Figure 5) is constructed on a concrete slab, and a shaft encoder and analog recorder are mounted to a steel table that is attached to the extensometer slab. A calibrated steel tape connects the recorder to the top of the inner pipe; because the steel table is anchored to the concrete slab, changes in land-surface altitude can be accurately measured and recorded. These recorded values through time represent the cumulative compaction that has occurred at the extensometer site. Because the extensometer functions as a piezometer and an extensometer, the cause and effect relation between the changes in water level in the aquifer and the changes in land-surface elevation can be established.

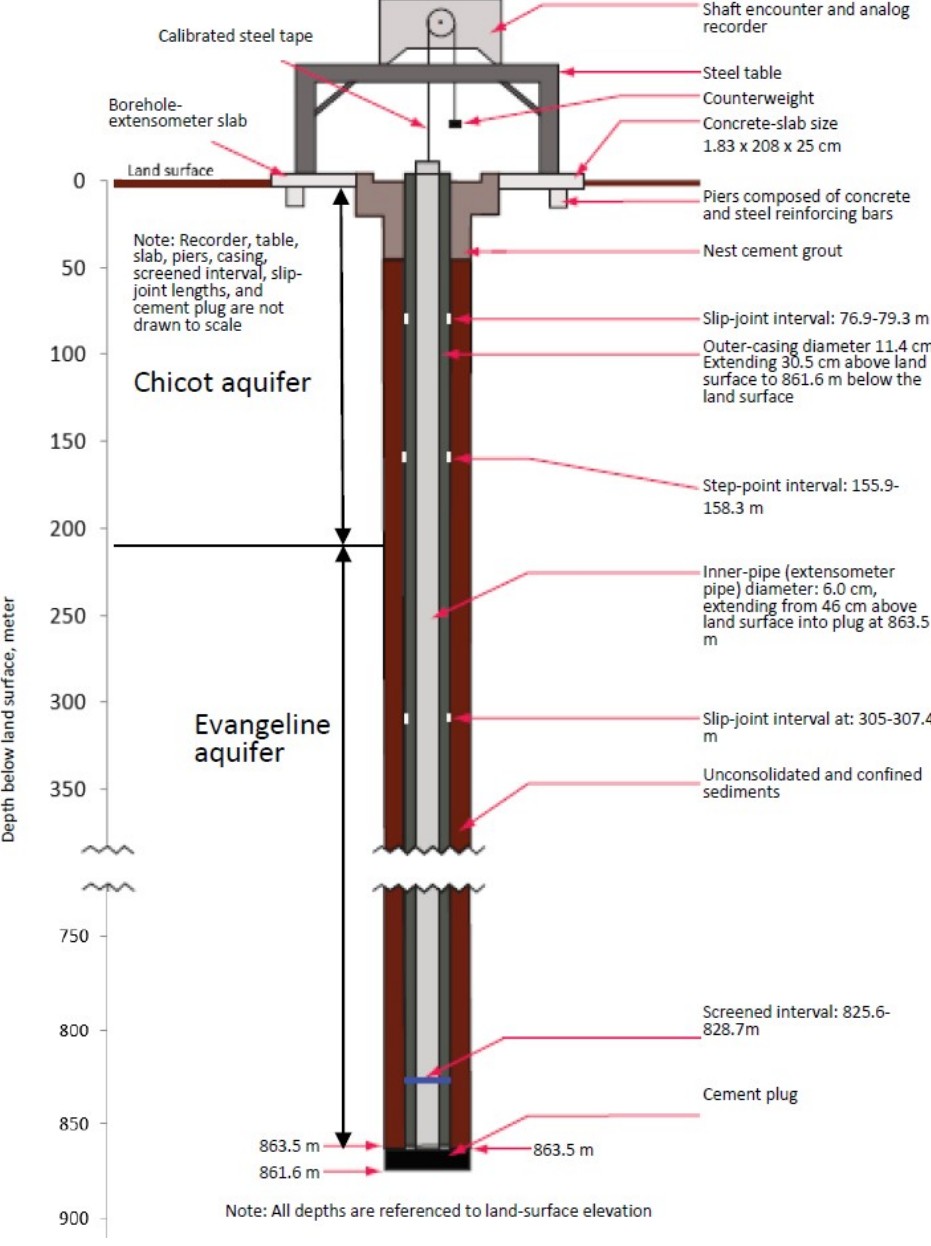

**Figure 5.** Example of cross-sectional perspective of the borehole extensometer/piezometer located at Pasadena, Texas (modified from [15]). (Map source: USGS [15]).

Borehole extensometer compaction data from the 11 sites (see Figure 4) are used to qualify compaction type and/or quantify the annual rate of compaction in the Chicot and Evangeline aquifers, thereby supporting groundwater level change management on control of land subsidence rates caused by changes in the amount of groundwater withdrawn from the Chicot and Evangeline aquifers. Five extensometers were installed in Harris (4) and Galveston (1) Counties and began recording compaction data in July 1973: East End, Baytown Shallow, Baytown Deep, and Seabrook in Harris County and Texas City in Galveston County (see Figure 4). The extensometer at Johnson Space Center had been installed in 1962 in Harris County but began recording compaction data in July 1973. Additional extensometers were added to the network during 1974–1976 in Harris County: Addicks in 1974, Pasadena in 1975, and Clear Lake Shallow and Clear Lake Deep in 1976 (see Figure 4). The final three extensometers were installed in Harris County in 1980: Lake Houston, Northeast, and Southwest (see Figure 4). Since activation or installation between 1973 and 1980, compaction data have been constantly recorded and periodically collected about every 28 days at the 13 extensometers on a routine basis, thereby providing site-specific rates of compaction accurate to within 0.3 mm [15].

## 4. Analysis Methodology

A pseudo-constant rate characteristic of secondary consolidation (creep) in compressible aquifer systems due to geo-historical overburden pressure is proposed with a new equation in this section after the variable rate characteristics of primary elastic and inelastic consolidation due to groundwater withdrawal are discussed. Then a combination equation of the pseudo-constant creep rate and the two variable rates of primary compaction is suggested to analyze an observed bulk compaction of compressible aquifer systems in response to groundwater level changes.

Almost all the permanent subsidence of a compressible aquifer system occurs due to the irreversible (or inelastic/nonrecoverable) compression or consolidation of aquitards through a slow (delayed) process of aquitard drainage [19]. This concept, which was labeled "the aquitard drainage model" by Helm [20], has formed the theoretical basis of many successful subsidence investigations [19,21–34]. The relation between changes in groundwater levels and compression of the aquifer system is based on the principle of effective stress first proposed by Karl Terzaghi [35]. By this principle, when groundwater level decreases due to discharges from an aquifer system, under a constant total load the support previously provided by the pore-fluid pressure will be transferred to the skeleton of the aquitard. Namely, the change in pore fluid pressure will be converted to effective stress on the skeleton of aquitards, which in turn causes the aquifer system's compaction. If the current effective stress is larger than the preconsolidation stress, the compaction is nonrecoverable (inelastic). In contrast, if effective stress is less than the preconsolidation stress, the compaction is recoverable (elastic) [1,24,30]. Conversely, when ground water level increases due to recharges to the aquifer system, the support previously provided by the skeleton reduces and the change in effective stress is shifted onto the pore fluid, which results in the aquifer system's elastic expansion. [1,24,30]. Therefore, in a general case the primary consolidation or compaction ($s_p$) can consist of two components: inelastic compaction ($s_{p-v}$) related to nonrecoverable specific skeletal storage ($S_{skv}$) of aquitard(s) or confining unit(s) and elastic compaction ($s_{p-e}$) associated with recoverable specific skeletal storage ($S_{ske}$) of aquitard(s) or confining unit(s) and sand layers in one aquifer system. Therefore, we have the following Equation (1)

$$s_p = s_{p-v} + s_{p-e} \tag{1}$$

The primary compaction rate can be described by Equation (2) from Equation (1)

$$\dot{s}_p = \dot{s}_{p-v} + \dot{s}_{p-e} \tag{2}$$

The $S_{skv}$ values are two magnitude orders larger than the values of ($S_{ske}$) [29,30,36–41]. This leads to inelastic compaction dominating land subsidence when it happens. Based on Terzaghi's consolidation theory [42], both $\dot{s}_{p-v}$ and $\dot{s}_{p-e}$ can be considered to be zero approximately when their consolidation degrees reach 99.4% while time factor $T_v$ (= $\Delta t/\tau'_0$, where $\Delta t$ is real time [T] and $\tau'_0$ is Terzaghi's time constant) equals 2.

Because of the weight of the overburden and the inelastic compaction characteristics of the clay layers, about 90 percent of the compaction is permanent [43]. Three main sedimentation stages are defined with respect to the concentration degree in self-weight consolidation as: the clarification regime, zone-settling regime, and compression regime [44]. The above Quaternary and Tertiary aquifer systems are still in the third compression stage. This compression was called as secondary consolidation (creep) by Taylor (1942) [45] or as "self-weight consolidation" by Been and Sills (1981) [46]. Therefore, it is assumed in this paper that secondary consolidation (creep) exists in the above unconsolidated aquifer systems due to geo-historical overburden pressure. For an unconsolidated sediment layer with an initial thickness of $H$ [L], the secondary consolidation $s_{s(t)}$ can be approximated by Equation (3) [45]

$$s_{s(t)} = C_\alpha H \log\left(\frac{t}{t_1}\right) \tag{3}$$

where $C_\alpha$ is the dimensionless coefficient of secondary compression of the sediment layer, $t_1$ is an initial reference time for secondary compression, $t$ is time larger than or equal to $t_1$. $\dot{s}_{s(t)} = (C_\alpha H/\ln 10)\frac{1}{t}$ follows from Equation (3) by taking the derivative with respect to time $t$ for subsidence rate. The decrease percentage ($D_S$) of $\dot{s}_{s(t)}$ from $t$ to $t + \Delta t$ can be derived by the authors with $D_S = 100[\dot{s}_{s(t)} - \dot{s}_{s(t+\Delta t)}]/\dot{s}_{s(t)}$ as

$$D_{s(t)} = \left(1 - \frac{t}{t + \Delta t}\right) \times 100 \tag{4}$$

$D_s$ approaches zero when $t \gg \Delta t$, which implies that $\dot{s}_s \approx$ *a constant*. In other words, the changing value of $\dot{s}_s$ over the $\Delta t$ period is difficult to be identified and can be ignored. This invariable rate is called a pseudo-constant rate of secondary consolidation in this paper. Table 1 shows how many years are needed for three different decrease percentages (1.0, 0.5 and 0.1%) of the specified subsidence rates in one given period. For example, if a period ($\Delta t$) is considered to be 10 years, 990, 1990, and 9990 years needed for specified subsidence rate decrease percentages of 1.0, 0.5, and 0.1%, respectively. The secondary consolidation rate $\dot{s}_s$ is a pseudo-constant if 1.0, 0.5, and 0.1% subsidence rate changes are negligible. The secondary consolidation for the Quaternary and Tertiary sediments can be considered to have been more than 1000 yrs. since the youngest and uppermost sediments of the Holocene Chicot aquifer were formed in the Greenlandian Age (4200 to 8200 years ago) and the Northgrippian Age (8200 to 11,700 years ago).

Therefore a bulk subsidence rate $\dot{s}_{(t)}$ can be the sum of primary inelastic compaction rate $\dot{s}_{p-v(t)}$, primary elastic compaction rate $\dot{s}_{p-e(t)}$ and secondary compaction rate $\dot{s}_{s(t)}$, i.e.,

$$\dot{s}_{(t)} = \dot{s}_{p-v(t)} + \dot{s}_{p-e(t)} + \dot{s}_{s(t)} \tag{5}$$

Equation (5), which is suggested by the authors in this paper, is employed to analyze extensometer measured compaction rate for the three components in response to groundwater level changes in aquifers. Three distinct compaction characteristics for the three components must be correctly applied in this analysis: inelastic compaction rate $\dot{s}_{p-v(t)}$ is 10 to over 100 times larger than elastic compaction rate $\dot{s}_{p-e(t)}$ when groundwater level is lower than preconsolidation pressure head; elastic compaction rate $\dot{s}_{p-e(t)}$ can be negative (land rebounding) while inelastic compaction rate $\dot{s}_{p-v(t)}$ decreases rapidly but never negative when groundwater is recovering; and secondary compaction rate $\dot{s}_{s(t)}$ does not change with groundwater level.

**Table 1.** Time of the secondary consolidation needed for specified subsidence rate decrease in given periods.

| $D_S$ [1] | Given Time Period $\Delta t$ in Equation (4), Years | | | | | |
|---|---|---|---|---|---|---|
| | 5 | 10 | 20 | 30 | 40 | 50 |
| 1.0% | 495 [2] | 990 | 1980 | 2970 | 3960 | 4950 |
| 0.5% | 995 | 1990 | 3980 | 5970 | 7960 | 9950 |
| 0.1% | 4995 | 9990 | 19,980 | 29,970 | 39,960 | 49,950 |

[1]: $D_S$: the decrease percentage of $\dot{s}_s$ in Equation (4); and [2]: The subsidence rate change is 1.0% for a 5-year period when the secondary consolidation elapses 495 years.

## 5. Results

In this section, groundwater withdrawal historical characteristics are firstly outlined in four different periods before the temporal and spatial characteristics of groundwater level change are summarized corresponding to the four periods, respectively. Then the analysis methodology in Section 4 is applied to analyzing the starting and ending of primary consolidation and emerging of creep consolidation based on the observed relationship between groundwater level change and compaction, which leads findings of bulk pseudo-constant secondary consolidation rate value at each location of the 13 extensometers.

### 5.1. Groundwater Withdrawal

Artificial primary consolidation first occurred in the early 1900s in areas where ground water, oil, and gas were extracted. It continued throughout the 20th century due primarily to groundwater pumpage [1,6]. Groundwater withdrawal in the Houston-Galveston region experienced the following four periods (see Figure 6c) [6]: 1) Slight withdrawal of about 0.19 million m$^3$/day from 1891 to 1930 for 40 years; 2) Increasing withdrawal rates for 46 years from 0.57 million m$^3$/day in 1931 to 4.28 million m$^3$/day in 1976 with an average growth of 0.05 million m$^3$/day per year. Near Texas City the withdrawal of ground water for public supply and industry caused more than 0.5 m of subsidence between 1906 and 1943 [1]; 3) Decreasing withdrawal rates for 25 years from 4.28 million m$^3$/day in 1976 to 3.03 million m$^3$/day in 2001; and 4) Roughly stable withdrawal rates of around 3 million m$^3$/day from 2001 to current for more than 17 years.

### 5.2. Groundwater Level Changes

The lowering of groundwater level in the Houston-Galveston region due to groundwater withdrawal in Section 5.1 was observed by the USGS and simulated by them with MODFLOW [6]. Based on simulated results in Figure 6a,b, from 1891 to 1900 the groundwater levels were about 21.35 and 9.15 m in the Chicot and Evangeline aquifers, respectively. This would be close to the status in the predevelopment of groundwater before 1891. During 1901 to 1930 they were lowered to 8.2 and 5.2 m in the Chicot and Evangeline aquifers, respectively. By 1937, groundwater levels were falling in a growing set of gradually coalescing cones of depression centered on the areas of heavy use [1]. Based on measured results in Figure 6a,b, in 1977 the lowest groundwater levels were −2 and −86 m in the Chicot and Evangeline aquifers, respectively; then groundwater levels were raised about 41.5 m to −40 m in 2008 for Chicot aquifer and 43 m to −43 ft in 2008 for Evangeline aquifer. After 2008, groundwater levels have been roughly stable.

Figure 7a,b show the lowest groundwater level depression cones for Chicot and Evangeline aquifers, respectively, which were found by USGS Subsidence Reviewer. Through checking groundwater level data monitored and published by USGS, the lowest groundwater levels were −97.1 m on 1/12/1990 at Well LJ-65-21-150 for the Chicot aquifer and −125.0 m on 1/9/1984 at Well LJ-65-13-904 for the Evangeline Aquifer, respectively. The maximum drawdown caused by groundwater withdrawal was estimated to be about 112 m in 1990 for the Chicot aquifer and 134 m in 1984 for the Evangeline aquifer, respectively, in the Houston-Galveston region. Based on the two wells, groundwater levels were raised about 54.6 m to −42.4 m in 2010 for the Chicot aquifer and 64.4 m to −56.0 m in 2005 for the Evangeline aquifer. Chico aquifer and Evangeline aquifer groundwater levels were roughly stable after 2010 and

2005, respectively, at the two wells. The subsidence bowl in Figure 1 is consistent with the groundwater level depression cones in Figure 7a,b.

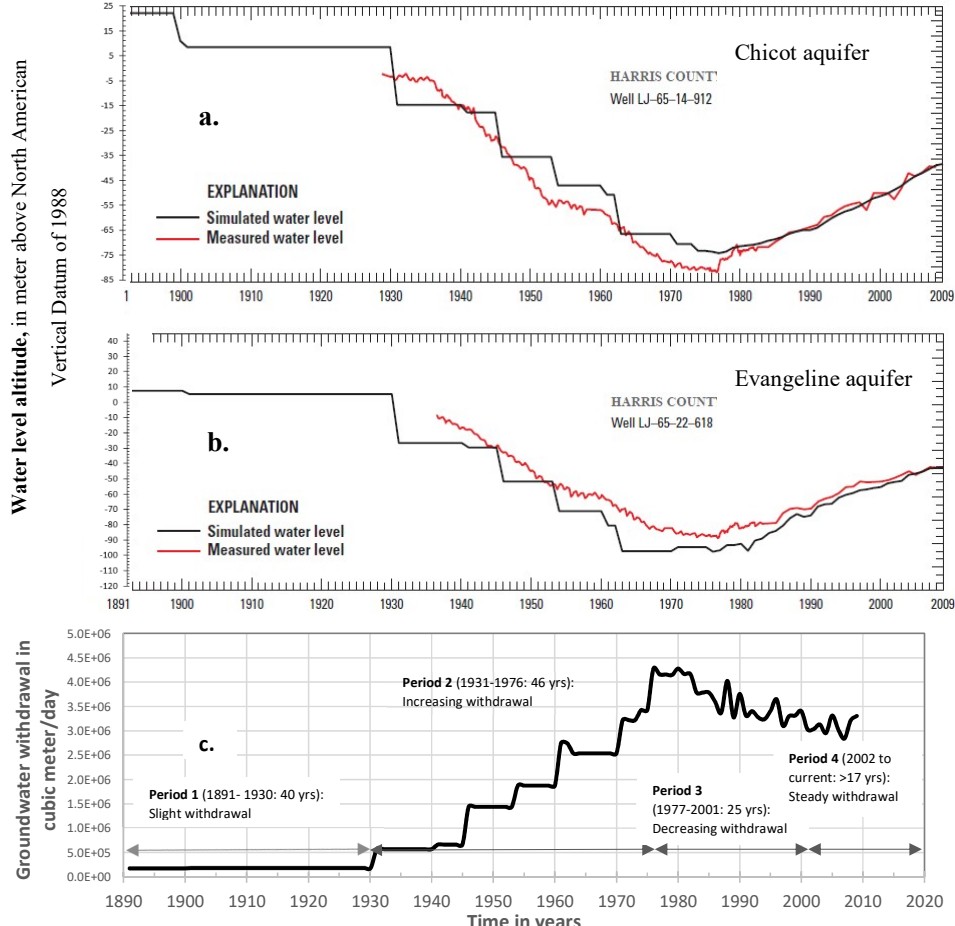

**Figure 6.** Groundwater withdrawal and ground water level fluctuations in the Houston-Galveston region: (**a**) Measured and simulated groundwater level in the Chicot aquifer (after [6]); (**b**) Measured and simulated groundwater level in Evangeline Aquifer (after [6]); and (**c**) Groundwater withdrawal (data from [6]). The location of water level well in Figure 6a,b can be find in Figure 4.

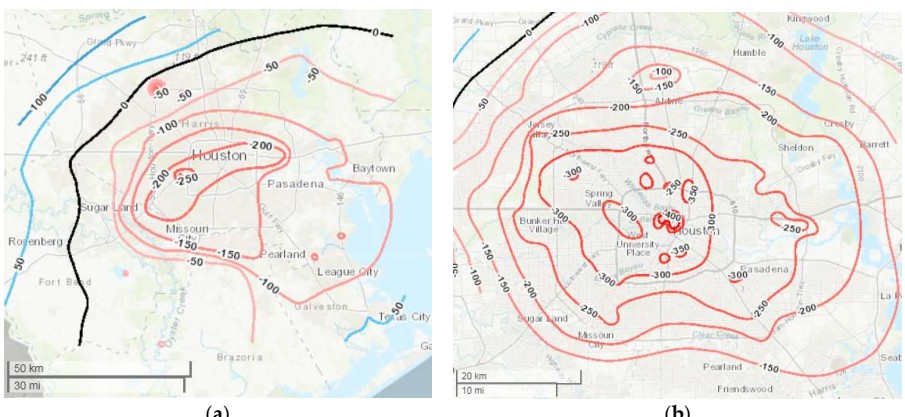

**Figure 7.** Measured groundwater level depression cones (in ft (m): 100 (30.5), 50 (15.25), −50 (−15.25), −100 (−30.5), −150 (−45.75), −200 (−61.00), −250 (−76.25), −300 (−91.5), −350 (−106.75), −400 (−122.00)) (**a**) in Chicot aquifer (lowest: −318.5 ft (−97.14 m) on 1/12/1990); and (**b**) in Evangeline aquifer (lowest: −409.98 ft (−125.04 m) on 1/9/1984) (from USGS Subsidence Viewer on https://txpub.usgs.gov/houston_subsidence/home/).

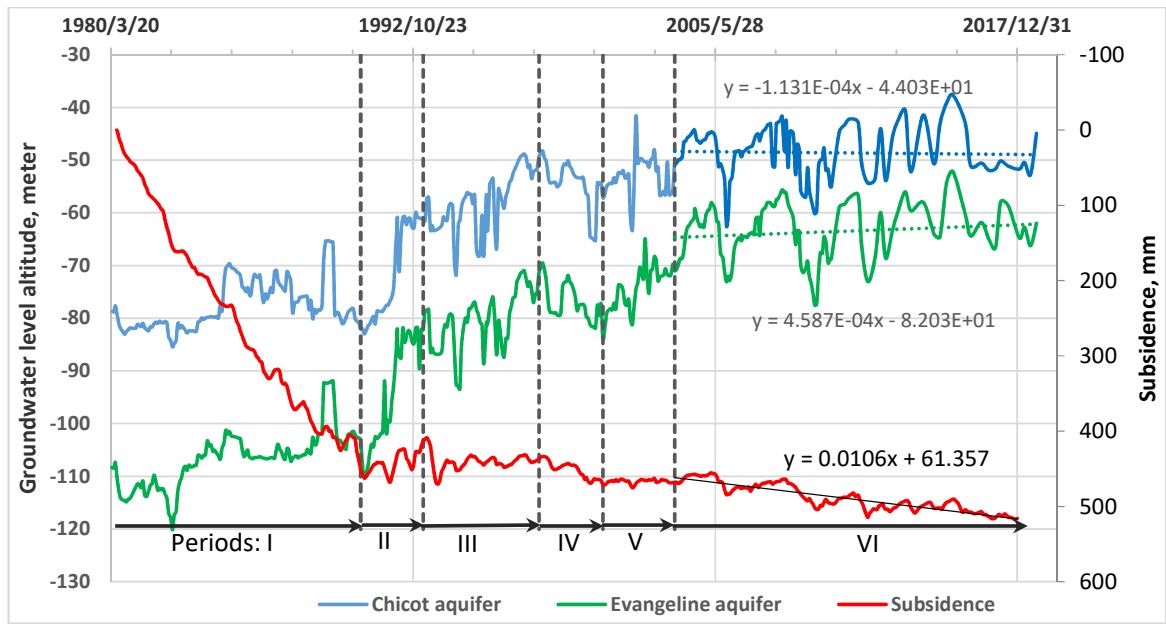

**Figure 8.** Groundwater levels in Chicot and Evangeline aquifers and land subsidence at extensometer site Southwest in the Houston-Galveston region. Calculated consolidation rates corresponding to groundwater levels during the following periods I ($\sim 8/16/1990$) : $\dot{S}_{p-v} \gg \dot{S}_{p-e} + \dot{S}_s > 0$; II (8/16/1990 to 3/25/1993): $\dot{S}_{p-e} < 0$ (rebounding), $\dot{S}_{p-v} + \dot{S}_s > 0$ and $\dot{S} = \dot{S}_{p-v} + \dot{S}_{p-e} + \dot{S}_s < 0$ (rebounding from Equation (3)); III (3/25/1993 to 1/22/1998): $\dot{S}_{p-e} < 0$, $\dot{S}_{p-v} + \dot{S}_s > 0$ and $\dot{S} = \dot{S}_{p-v} + \dot{S}_{p-e} + \dot{S}_s \approx 0$; IV (1/22/1998 to 9/20/2000): $\dot{S}_{p-e} > 0$, $\dot{S}_{p-v} \rightarrow 0$ and $\dot{S} = \dot{S}_{p-v} + \dot{S}_{p-e} + \dot{S}_s > 0$; V (9/20/2000 to 9/18/2003): $\dot{S}_{p-e} < 0$, $\dot{S}_{p-v} \approx 0$ and $\dot{S} = \dot{S}_{p-e} + \dot{S}_s \approx 0$; and VI (9/18/2003 to 12/31/2017): $\dot{S}_{p-e} \approx 0$, $\dot{S}_{p-v} \approx 0$ and $\dot{S} = \dot{S}_{p-v} + \dot{S}_{p-e} + \dot{S}_s \approx \dot{S}_s$. The slope values in the trend equations for groundwater levels are expressed in m/day and the slope value in the trend equation for subsidence is in mm/day.

### 5.3. End of Primary Inelastic Compaction

It will be firstly demonstrated in detail in this section how Equation (5) is employed to correctly analyze inelastic, elastic, and secondary (creep) compaction characteristics in six different periods of groundwater level changes in the Chicot and Evangeline aquifers from 1980 to 2017 from the total compaction measurements of the two aquifer systems at extensometer site Southwest in the Houston-Galveston region (Figure 8). Then this method is applied to other 12 extensomter sites.

Only the Chicot aquifer and Evangeline aquifers are involved with the 12 extensometers except Baytown Deep at Baytown (see Figure 4). From Table 2, a total of 19 groundwater level wells in the two aquifers at or near the 11 extensometer station locations were employed in this paper to analyze the change of inelastic and elastic compaction and secondary consolidation in response to trends of groundwater level fluctuation based on the methodology in Section 4 (Equation (5)). As one example, Figure 8 shows the results based on analysis of observed land subsidence at borehole extensometer Southwest (Figure 4) from 17 June 1980 to 28 December 2017 and monitored groundwater levels at Chicot aquifer Well LJ-65-21-229 and Evangeline aquifer Well LJ-65-21-227 from 4 April 1980 to 1 October 2018. The preconsolidation hydraulic head was set to be −21.35 m in HAGM model [6] for the aquitards within the two aquifers. Six periods were divided based on variable rate characteristics of elastic, inelastic, and creep compaction corresponding to groundwater level change. From Figure 8 during Period I (4/4/1980~ 8/16/1990), Groundwater levels in the Chicot and Evangeline aquifers changed from −65 to −85 m and from −92 to −121 m, respectively. Both remained much lower than the −21.35 m of the initial preconsolidation pressure head. Thus, the inelastic compaction from the aquitards dominated the subsidence at this location: $\dot{S}_{(t)}$ was 46.92 mm/yr from 1980 to 1987 then decreased to 31.46 mm/yr from 1988 to 1990. The subsidence characteristics during Period I would be

$\dot{s}_{p-v} \gg \dot{s}_{p-e} + \dot{s}_s > 0$. During Period II (8/16/1990 to 3/25/1993), groundwater levels in the Chicot and Evangeline aquifers were raised from −83 to −60 m and from −118 to −82 m, respectively. The 23 m and 36 m groundwater level rise caused a land rebounding rate of 14.9 mm/yr. Thus, elastic rebounding of the two aquifers dominated the deformation at this location. The subsidence characteristics during Period II would be $\dot{s}_{p-e} < 0$, $\dot{s}_{p-v} + \dot{s}_s > 0$ and $\dot{s} = \dot{s}_{p-v} + \dot{s}_{p-e} + \dot{s}_s < 0$. During Period III (3/25/1993 to 1/22/1998), groundwater levels in the Chicot and Evangeline aquifers were further raised about 12 m to reach −51 m and about 9 m to −72 m, respectively. The further 9 to 12 m trend in groundwater level recovery did not cause further land rebounding although the elastic compaction rate $\dot{S}_{p-e}$ was less than zero. The trend in the subsidence rate approached approximately zero, which implies $\dot{s}_{p-v} + \dot{s}_s > 0$ and $\dot{s}_{p-v} + \dot{s}_s \approx -\dot{s}_{p-e}$ from Equation (5). Thus, the elastic rebounding of the two aquifers approximately offset the combination of inelastic compaction and secondary consolidation at this location. The subsidence characteristics during Period III would be $\dot{s}_{p-e} < 0$, $\dot{s}_{p-v} + \dot{s}_s > 0$ and $\dot{s} = \dot{s}_{p-v} + \dot{s}_{p-e} + \dot{s}_s \approx 0$. During Period IV (1/22/1998 to 9/20/2000), groundwater levels in the Chicot and Evangeline aquifers were lowered about 17 m to −65 m and about 13 m to −82 m, respectively. The 13 to 17 m groundwater level lowering caused land subsidence in trend and the elastic compaction rate $\dot{s}_{p-e}$ is larger than zero. The inelastic consolidation from aquitards within the two aquifers continued for more than about 21 yrs with a decreasing rate $\dot{s}_{p-v}$, which approached zero $\left(\dot{s}_{p-v} \rightarrow 0\right)$ within this period, since the regional lowest groundwater levels happened due to the maximum groundwater withdrawal during 1977 to 1984. The subsidence characteristics in Period IV would be $\dot{s}_{p-e} > 0$, $\dot{s}_{p-v} \rightarrow 0$ and $\dot{s} = \dot{s}_{p-v} + \dot{s}_{p-e} + \dot{s}_s > 0$. During Period V (9/20/2000 to 9/18/2003), groundwater levels in the Chicot and Evangeline aquifers were raised again about 5 m to −50 m and about 10 m to −70 m, respectively. The 5 to 10 m groundwater level rise caused neither further land rebounding nor significant subsidence. This happened only when inelastic compaction ceased $\left(\dot{s}_{p-v} \approx 0\right)$ and when elastic rebounding offset the secondary consolidation ($\dot{s}_s \approx -\dot{s}_{p-e}$). Thus, it would appear that the delay in compaction from inelastic specific skeletal storage of aquitards within the Chicot and Evangeline aquifers at extensometer site Southwest ceased during or before 2000. The subsidence characteristics in Period V would be $\dot{s}_{p-e} < 0$, $\dot{s}_{p-v} \approx 0$ and $\dot{s} = \dot{s}_{p-e} + \dot{s}_s \approx 0$. During the last Period VI (9/18/2003 to 12/28/2017), groundwater levels in the Chicot and Evangeline aquifers exhibited an almost stable trend of $1.13 \times 10^{-4}$ m/day (Figure 8) (0.03 m/yr, Table 2) and $4.59 \times 10^{-4}$ m/day (Figure 8) (0.14 m/yr, Table 2), respectively. This leads to the conclusion that the trend in elastic compaction can be considered negligible $\left(\dot{s}_{p-e} \approx 0\right)$. Only secondary consolidation emerged ($\dot{s}_s > 0$) since both $\dot{s}_{p-e} \approx 0$ and $\dot{s}_{p-e} \approx 0$. Thus the subsidence characteristics in Period VI would be $\dot{s}_{p-e} \approx 0$, $\dot{s}_{p-v} \approx 0$ and $\dot{s} = \dot{s}_{p-v} + \dot{s}_{p-e} + \dot{s}_s \approx \dot{s}_s = 0.0106$ mm/day (3.87 mm/yr) (Figure 8).

The above analysis was applied to 10 other extensometer sites: Texas City, Seabrook, Johnson Space Center and Clear Lake (which share two of the same groundwater level wells), Baytown, Addicks, East End, Northeast, Pasadena, and Lake Houston with monitored groundwater level data from wells at or near the sites. Columns 5 and 6 in Table 2 give the approximate starting and ending dates, respectively, of the appearance of secondary consolidation at each extensometer site. The slope of the groundwater level trend is given in Column 7 in m/day and in Column 8 in m/yr during the appearance period at each site. All other 10 starting dates are after the starting date 9/18/2003 of the appearance of secondary consolidation at extensometer site Southwest.

The identified periods of the appearance of secondary consolidation for two extensometer groups (Baytown and Clear Lake and Space Center), in which each group shares same groundwater level monitoring wells, and another eight individual extensometers including Southwest are given in Table 2. All the periods are after the inelastic compaction ceased within Period V from 9/20/2000 to 9/18/2003 identified from Figure 8. The groundwater level trend values from 0.21 to 0.39 m/yr are a little bit large for extensometer sites East End, Northeast and Pasadena, but the groundwater level difference between the starting date and the ending date of the corresponding secondary consolidation appearance period is very small (0.02 to 0.14 m). This small groundwater level difference means the cumulative elastic deformation approached zero during the period.

**Table 2.** Secondary consolidation appearance periods based on groundwater level trend at or near extensometer sites in the Houston-Galveston region.

| Extensometer | Depth, m | Well # | Aquifer | Secondary Consolidation Appearance Period | | Groundwater Level Trend | | Secondary Consolidation Rate $\dot{s}_s$ | |
|---|---|---|---|---|---|---|---|---|---|
| | | | | Starting Date | Ending Date | m/day * | m/yr | mm/day | mm/yr |
| Texas City | 224 | KH-64-33-901 | Chicot | 1/24/2008 | 1/14/2017 | $-8.73 \times 10^{-5}$ | −0.03 | $2.222 \times 10^{-4}$ | 0.08 |
| Seabrook | 863 | LJ-65-32-519 LJ-65-32-630 | Chicot Evangeline | 1/25/2008 | 1/14/2017 | $-2.77 \times 10^{-5}$ $-7.67 \times 10^{-5}$ | −0.01 −0.03 | $8.317 \times 10^{-3}$ | 3.04 |
| Space Center Clear Lake Deep | 223 936 * | LJ-65-42-422 LJ-65-42-424 | Chicot Evangeline | 1/25/2007 | 12/20/2017 | $9.09 \times 10^{-5}$ $6.79 \times 10^{-5}$ | 0.03 0.02 | $5.062 \times 10^{-3}$ $3.033 \times 10^{-3}$ | 1.85 1.11 |
| Baytown Shallow Baytown Deep | 131 1475 | LJ-65-16-933 LJ-65-16-931 | Chicot Evangeline | 5/26/2005 1/11/2007 | 5/28/2009 | $3.82 \times 10^{-4}$ $9.21 \times 10^{-5}$ | 0.14 0.03 | $3.630 \times 10^{-3}$ $5.956 \times 10^{-3}$ | 1.33 2.17 |
| Addicks | 550 | LJ-65-12-729 LJ-65-12-726 | Chicot Evangeline | 10/1/2007 | 5/15/2014 | $2.73 \times 10^{-4}$ 0.00 | 0.10 0.00 | $2.327 \times 10^{-2}$ | 8.49 |
| East End | 421 | LJ-65-22-623 LJ-65-22-622 | Chicot Evangeline | 7/27/2007 | 1/13/2015 | $-2.72 \times 10^{-4}$ $5.64 \times 10^{-4}$ | −0.10 0.21 ^ | $4.926 \times 10^{-3}$ | 1.80 |
| Northeast | 550 | LJ-65-14-745 LJ-65-14-746 | Chicot Evangeline | 1/4/2008 | 3/1/2011 | $5.27 \times 10^{-4}$ $1.07 \times 10^{-3}$ | 0.19 0.39 ^^ | $1.150 \times 10^{-2}$ | 4.20 |
| Pasadena | 863 | LJ-65-23-321 LJ-65-23-326 | Chicot Evangeline | 1/5/2007 2/6/2007 | 1/5/2011 3/30/2010 | $3.76 \times 10\text{-}6$ $8.39 \times 10^{-4}$ | 0.00 0.31 ~ | $6.032 \times 10^{-3}$ | 2.20 |
| Lake Houston | 719 | LJ-65-07-902 LJ-65-07-908 | Chicot Evangeline | 1/7/2004 | 4/4/2007 | $8.38 \times 10^{-5}$ $1.24 \times 10^{-4}$ | 0.03 0.05 | $3.760 \times 10^{-3}$ | 1.37 |
| Southwest | 719 | LJ-65-21-229 LJ-65-21-227 | Chicot Evangeline | 9/18/2003 | 10/18/2018 | $-1.13 \times 10^{-4}$ $4.59 \times 10^{-4}$ | −0.04 0.17 | $1.042 \times 10^{-2}$ | 3.80 |

*: Clear Lake Deep is 936 m deep and Clear Lake Shallow is 226 m deep. ^: The difference of groundwater level between −42.75 m on 5/7/2008 and −42.77 m on 11/3/2014 is 0.02 m. ^^: The difference of groundwater level between −54.14 m on 1/4/2008 and −54.18 m on 3/1/2011 is 0.04 m. ~: The difference of groundwater level between −40.03 m on 2/6/2007 and −40.17 m on 3/30/2010 is 0.14 m.

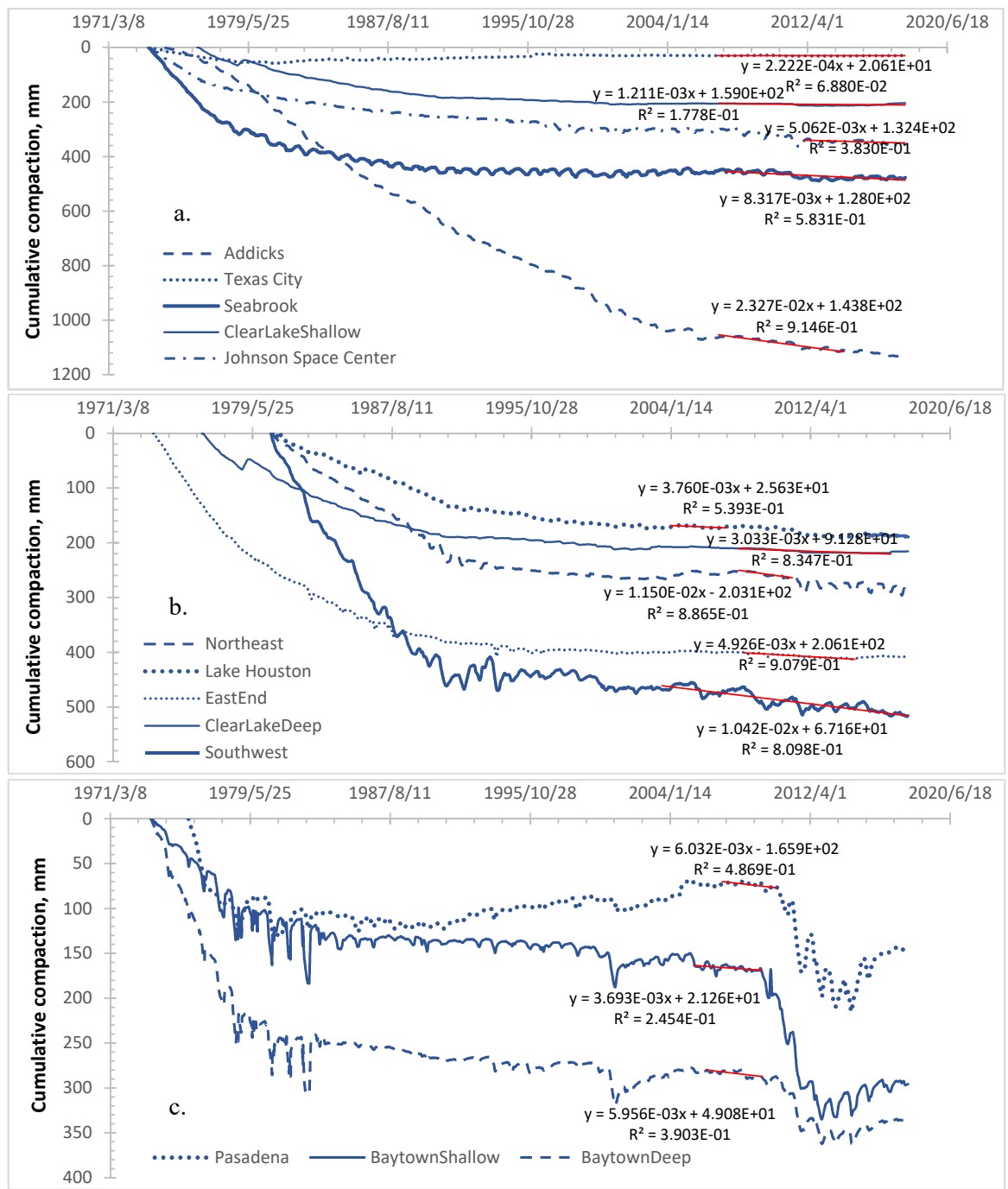

**Figure 9.** Linear trends of secondary consolidation from observed cumulative compactions at 13 extensometer locations in the Houston-Galveston region (data source: USGS). The secondary consolidation period for each site is given in Table 2. The slope values in the trend equations are in mm/day. (**a**): 5 extensometers Addicks, Texas City, Seabrook, ClearLakeShallow, and Johnson Space Center; (**b**): 5 extensometers Northeast, Lake Houston, EastEnd, ClearLakeDeep, and Southwest; and (**c**): 3 extensometers: Pasadena, BaytownShallow, and BaytonDeep.

Figure 9 shows the measured cumulative subsidence curves with time at the 13 extensometer sites in the Houston-Galveston region. The secondary consolidation trend line and equation during stable groundwater level periods identified in Table 2 are shown for each subsidence curve in Figure 9. The reason for the huge subsidence fluctuations at Baytown Shallow and Deep and Pasadena from 2010 to 2014 (see Figure 9c) has not previously been well understood based on Kasmarek et al. (2016) [7]

and the author's (Y. Liu) personal discussion with USGS hydrologist, Jason Ramage. Ramage wrote in his email that water usage in the area has not increased. The only other physical factor he could postulate is reaction along faults in the area, which was detected by mapping ground deformation with multi-temporal InSAR [4]. However, if this were the case, he would still think all extensometer locations would be affected similarly. All other sites present slightly or little increased subsidence rate over this period of 2010 to 2014 (Figure 9a,b) as pointed out by Kasmarek (2016) [7], but it still seems apparent that secondary compaction at all other 10 extensometer sites has likely dominated land subsidence without primary inelastic consolidation occurring since around 2003. Although secondary consolidation will continue for a very long period into future, inelastic and elastic compaction has fully and successfully been placed under control in the Houston-Galveston region since 2003. This achievement, the controlled land subsidence due to groundwater withdrawal, contributes to a long-term groundwater level change management from 1977 to 2002. This groundwater level change management recovered the lowest groundwater levels in the Chicot and Evangeline aquifers produced by a high groundwater withdrawal of about 4.3 million $m^3$/day during 1977 and 1984 to expected high stable groundwater levels in the two aquifers by decreasing that withdrawal to about 3.0 million $m^3$/day since around 2002.

The temporal variation of the pseudo-constant secondary consolidation could not be identified well from the current compaction observations in the HGR because its emerging period is just 2 to 15 years from Figure 9 and Table 2. However, the secondary consolidation rate of 0.08 to 8.49 mm/yr spatially varies significantly with location. The reason for this spatial variation would include total thickness of compressible aquifer systems, clay cumulative thickness percentage, individual aquitard thickness [30], and overburden pressure history. For example, the reason for the minimum creep rate of 0.08 mm/yr at Texas City would be (1) the total compressible aquifer system is 244 m of the Chicot aquifer; (2) the percentage of clay cumulative thickness in the Chicot aquifer is about 20.8% [14], which is the minimum value at the 13 extensometer locations with average of 52.1%; (3) the equivalent thickness of aquitards [30] would be very thin. However, the creep rate of 1.33 mm/yr at Baytown Shallow is 16.6 times larger than 0.08 mm/yr at Texas City probably because the clay percentage is 50.0% in the 131 m thick Chicot aquifer, its equivalent aquitard thickness would be larger, and the Chicot aquifer has the shortest overburden pressure history in the four compressible aquifer systems.

## 6. Conclusion

Based on the aquitard drainage model [20], primary consolidation (or compaction) consists of two components: inelastic compaction from nonrecoverable specific skeletal storage of aquitard(s) or confining unit(s) and elastic compaction from recoverable specific skeletal storage of aquitard(s) or confining unit(s) and sand layers in the aquifer system.

After primary inelastic consolidation is completed and when primary elastic consolidation rate trends to zero, the remaining compaction of an unconsolidated or semi-consolidated aquifer system measured by a borehole extensometer is considered here to be the secondary (creep) consolidation due to geo-historical overburden pressure. Based on Taylor's (1942) secondary consolidation theory, the rate of secondary consolidation behaves like a pseudo-constant, especially if it has elapsed over 1000 years. Secondary consolidation within the Chicot and Evangeline aquifer system has likely existed for at least for 1000 years since the youngest and uppermost sediments of the Holocene Chicot aquifer were formed in the Greenlandian Age (4200 to 8200 years ago) and the Northgrippian Age (8200 to 11,700 years ago).

A bulk land subsidence rate is assumed to be the sum of inelastic and elastic compaction rates due to groundwater withdrawal (the traditional primary consolidation process of geotechnical engineers) and a secondary consolidation rate due to geo-historical overburden pressure. Primary compaction/consolidation can disappear under groundwater level recovery management, but secondary consolidation continues at a pseudo-constant rate during a long-term consolidation period such as over 1000 years.

As much as 3.05 m of land subsidence was measured in 1979 in the Houston-Galveston region. It was largely a result of inelastic/nonrecoverable compaction within the Chicot and Evangeline aquifers from 1937 to 1979 and reflected the continuous changes in the preconsolidation pressure head within the aquitards that lie within these two aquifer systems. All these changes were due in turn to groundwater levels lowering in response to a continuous increase in groundwater withdrawal rates from 0.57 million m$^3$/day to 4.28 million m$^3$/day. Groundwater level recovery management from 1979 to 2000 successfully decreased inelastic compaction from ~40 mm/yr to zero by decreasing groundwater withdrawal rates from 4.3 million m$^3$/day to 3.0. A little rebounding of the land surfaces was achieved from this management of groundwater levels. An additional pseudo-constant secondary consolidation rate of 0.08 to 8.49 mm/yr emerged from field extensometer data as a result of the analysis described in the present study. This additional skeletal compression occurred while the groundwater levels in the two aquifers were being managed. The trends in this secondary compression have tended to remain stable since 2000. The secondary consolidation process would be beyond any current groundwater level change management scheme because it is likely caused by geo-historical overburden pressure upon and within the two aquifers.

Borehole extensometer compaction measurements not only successfully corroborate the efficacy of the groundwater level change management process for controlling land subsidence, but they also reveal for the first-time secondary consolidation subsidence in the Quaternary and Tertiary aquifer systems in the Houston-Galveston region.

It should be pointed out that a reason for the huge subsidence fluctuations at Baytown Shallow and Deep and at Pasadena from 2010 to 2014, which previously has not been well understood, would likely be not only primary but also secondary compaction.

**Author Contributions:** Conceptualization, Y.L.; Data curation, Y.L.; Formal analysis, Y.L.; Funding acquisition, J.L, Y.L. and Z.N.F.; Investigation, Y.L.; Methodology, Y.L.; Project administration, J.L., Y.L. and Z.N.F.; Supervision, J.L., Y.L. and Z.N.F.; Writing—original draft, Y.L.; Writing—review & editing, Y.L., J.L. and Z.N.F.

**Funding:** This research is supported by the National Science Foundation (NSF) grant no. 1832065 entitled "Identification of urban flood impacts caused by land subsidence and sea level rise in the Houston-Galveston region".

**Acknowledgments:** The authors thank NSF for supporting this research through grant no. 1832065. The authors express their gratitude to Jason Ramage for his help in data collection, explanation, and fixing and William Mike Chrismer for his help in data collection. The authors appreciate Donald C. Helm and four anonymous reviewers for their constructive comments for improving the manuscript. The compaction data and groundwater data are from USGS (https://txpub.usgs.gov/houston_subsidence/home/).

**Conflicts of Interest:** The authors declare no conflict of interest.

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
