# Peer review of "Groundwater Level Change Management on Control of Land Subsidence Supported by Borehole Extensometer Compaction Measurements in the Houston-Galveston Region, Texas"

_geosciences, doi:10.3390/geosciences9050223_

Round 1

Reviewer 1 Report

I think that the introduction lacks some key publications about land subisdence in the region of interest (i.e. Houston-Galveston).

It is mandatory to add the following reference Qu et al., 2015, Mapping ground deformation over Houston–Galveston, Texas using multi-temporal InSAR, Remote Sensing of Environment" and some other references therein. this isimportant to set up the context of land subsidence and the way it has been monitored and estimated very recently by new modern satellite technologies for instance.

Moreover, i'm quite sure that some ideas and observations are explained in that reference and should help to explain the specific subsidence pattern around Baytown and Pasadena that are just ruled out without trying to find the right answers.

So the introduction chapter need to be partly updated, the scientific issues about the specific pattern observed at two extensometer sites, the references need to be checked as well as they are not well fully written (missing journal, issue, etc) meaning that it is impossible for the reader to check them and of course some key references are missing as well.

Reviewer 2 Report

The manuscript describes how groundwater level changes relate to land subsidence in the Houston-Galveston Region, Texas. Although there are some interesting points, the manuscript lacks a rationale and motivation. Moreover, the authors talk about groundwater modeling, but they have not provided details to judge the quality of the simulations. In addition,  I did not find a clear relationship between the subsidence and water level drops. Although it is intuitive to think that these two would be related, results do not suggest a clear relationship. 

More specific comments:

(1) Figure 6 shows Groundwater withdrawal and groundwater levels fluctuations in the Houston-Galveston region. 

(a) It is not clear where (at what location) these water levels were measured and simulated. 

(b) These are large aquifers, so what was the spatial variability in water levels/withdrawal?

(c) It is also not shown in the paper what kind of parameters were chosen. 

(d) Was it a steady state simulation or transient? 

(e) Why simulations show a drop in water level like a step function?

(2) It is also not clear why the water level started increasing although the usage was restricted, but what was the source of the replenishment? Should not it remain stable?

(3) How about changes in recharge areas due to consecutions? Authors need to address these issues and provide a clear description of the model and parameters in the manuscript before the quality of simulations can be judged. 

(4) It is hard to say from Figure 8 whether groundwater levels fluctuations and subsidence have a clear relationship. Authors need to clarify this point in the introduction, methods, and results accordingly.

(5) Figure 9 – the authors mention the linear trend, but it looks more like an exponential decay. What are R2 values for linear vs. exponential functions?

Reviewer 3 Report

The concept of the manuscript is interest, however there are no innovative tools to render this work of international interest. Many maps should be revised adding coordinates, increase the quality etc. Additionally, critical hydrogeological information is missing  from the manuscript. 

Reviewer 4 Report

I carefully read the manuscript geosciences-427970, untitled ''Groundwater Level Change Management on Control of Land Subsidence Supported by Borehole Extensometer Compaction Measurements in the Houston-Galveston Region, Texas''. The paper shows a very interesting research about the secondary consolidation rate caused by geo-historical overburden pressure over Houston-Galveston Region through extensometer observations from 1980 to 2017. The paper utilized the measured cumulative subsidence at the 13 extensometer sites to analysis inelastic, elastic and secondary compaction characteristics in the Chicot and Evangeline aquifers, and captured the starting and ending dates, as well as the rate of secondary consolidation in the two aquifers. I found the paper generally well written, organized in good shape and structure. But my personal impression is that the innovative core of the paper is limited, most of results were cited from published reports, and the analysis of some parts was still not deep enough. So there are spaces for improvements of this paper. In the following I provide comments and suggestions for this manuscript.

1) In "Abstract", the authors talked too much on study background. Please simplify these kinds of contents and put more focus on the results and findings, specially address your innovative work.

2) We usually summarize the research in a small paragraph telling reader about what will the authors are going to discuss in each section.

3) Where are equations (4) and (5) from? Please state it out if you derive them by yourself, otherwise you should label references.

  In addition, I would recommend you label references in the main text although you have labeled them in the figure caption (e.g., section 5.1 and 5.2).  

4) Lines 324-325: Please mark well locations on Fig. 4.

5) In section 5.3, the authors divide the study period into seven intervals, so could you tell us according to what were the intervals determined?

6) Fig. 9: I would suggest you use only one blue color line in each subplot.

7) Lines 393-411: Significant increments of subsidence were observed at Baytown Shallow (about 0.11ft/y) and Deep (about 0.06ft/y) and Pasadena (about 0.1ft/y) during 2010 and 2014, which all located near the eastern Houston. However according to Kasmarek et al., (2017), all the other sites also present increased subsidence rates over this period though the variations are not significant as the Baytown Shallow and Deep and Pasadena sites (e.g., Seabrook, about 0.025ft/y; ). So it seem s to me all extensometer locations showed abnormal subsidence signals on form 2010 to 2014.

8) The estimated secondary consolidation rates vary spatially and temporally over Houston-Galveston region, what’s the reason for the phenomenon? I would suggest the authors present more discussion on this part.   

Typos: Line 361: one equation should be.

Author Response

Dear Reviewer,

Thank you very much for your comments. Please find attachment for our detailed responses.

Best regards,

All authors

Round 2

Reviewer 2 Report

Authors have not addressed comments adequately. The manuscript makes use of modeling performed by the USGS; however, authors should provide enough details to understand the quality of their work. Authors could provide details as Supplemental Material.

Reviewer 3 Report

The manuscript has been revised as much as possible. I think it can published in Geosciences journal.

Author Response

Thank you very much!

Reviewer 4 Report

Comments from the first submission are properly explained and revised in the revised manuscript. Even though the real innovative core of the methodology is limited, the authors presented very good technological application, and represented well explanations to the observed phenomena. Please pay attention to my comment 2 in the last round. The authors need only summarize the research in a small paragraph telling reader about what will the authors are going to discuss in each section as the last paragraph of Introduction part. 

Round 3

Reviewer 2 Report

The last time I made comments and they were not adequately addressed. Now authors have included some information as supplemental. I do not agree with authors about their answer regarding spatial and temporal distribution of wells from the first revision. They also refused to strengthen the rationale and motivation.
